# Evolution of Toughening Mechanisms in PH13-8Mo Stainless Steel during Aging Treatment

**DOI:** 10.3390/ma16103630

**Published:** 2023-05-10

**Authors:** Honglin Zhang, Peng Mi, Luhan Hao, Haichong Zhou, Wei Yan, Kuan Zhao, Bin Xu, Mingyue Sun

**Affiliations:** 1Key Laboratory of Nuclear Materials and Safety Assessment, Institute of Metal Research, Chinese Academy of Sciences, Shenyang 110016, China; 2Shenyang National Laboratory for Materials Science, Institute of Metal Research, Chinese Academy of Sciences, Shenyang 110016, China; 3China Aerodynamics Research and Development Center, Mianyang 621000, China; 4School of Mechanical Engineering, Yanshan University, Qinhuangdao 066004, China; 5Shi-Changxu Innovation Center for Advanced Materials, Institute of Metal Research, Shenyang 110016, China

**Keywords:** PH13-8Mo stainless steel, aging treatment, heterogeneous microstructures, toughening mechanisms

## Abstract

PH13-8Mo stainless steel has been widely used in aerospace, petroleum and marine construction, obtaining continuous investigation attention in recent years. Based on the response of a hierarchical martensite matrix and possible reversed austenite, a systematic investigation of the evolution of the toughening mechanisms in PH13-8Mo stainless steel as a function of aging temperature was carried out. It showed there was a desirable combination of high yield strength (~1.3 GPa) and V-notched impact toughness (~220 J) after aging between 540 and 550 °C. With the increase of aging temperature, the martensite matrix was recovered in terms of the refined sub-grains and higher ratio of high-angle grain boundaries (HAGBs). It should be noted there was a reversion of martensite to form austenite films subjected to aging above 540 °C; meanwhile, the NiAl precipitates maintained a well-coherent orientation with the matrix. Based on the post mortem analysis, there were three stages of the changing main toughening mechanisms: Stage I: low-temperature aging at around 510 °C, where the HAGBs contributed to the toughness by retarding the advance of cracks; Stage II: intermediate-temperature aging at around 540 °C, where the recovered laths embedded by soft austenite facilitated the improvement of toughness by synergistically increasing the advance path and blunting the crack tips; and Stage III: without the coarsening of NiAl precipitates around 560 °C, more inter-lath reversed austenite led to the optimum toughness, relying on “soft barrier” and transformation-induced plasticity (TRIP) effects.

## 1. Introduction

Due to its comprehensive properties of high strength, toughness, excellent corrosion resistance and processing characteristics, PH13-8Mo stainless steel has widely been used in aerospace, the nuclear industry and offshore platform construction [1,2,3], including components of landing gear, fastening bolts and pressure vessels. Although serving as a commercial precipitation-hardening steel, there has been a lot of research on precipitation modifications [4], behaviors of stress corrosion cracking [5] and hydrogen embrittlement [3] during the past decades. Nano-scale precipitation and its strengthening effect has achieved the most attention among them. Guo et al. [6] ascribed the initial hardening effect to the interaction between solute and dislocations; under an elevated aging temperature, the size of the B2-type NiAl phase became larger, yet without coarsening [7], due to the low diffusivity of Mo and Cr elements [8,9]. Assisted by atom probe tomography (APT), Leitner et al. [4] traced the precipitation sequence and evaluated the strengthening mechanisms using different models.

To the best of our knowledge, less attention has been paid to the toughening mechanisms of PH13-8Mo steel. It is known that the reversed/retained austenite is beneficial to the toughness due to “soft barrier” [10] and transformation-induced plasticity (TRIP) effects [11]. Schnitzer found that 30 vol.% of reversed austenite formed during prolonged aging at 575 °C [12], which led to an increased strain-hardening exponent and energy absorption [13]. However, the impact toughness has a nonlinear relationship with the amount of austenite in respect of its morphology [14] and chemical composition [15]. The formation of austenite decreases the yield strength of precipitation-hardening steel. For most cases, the toughening from the martensite matrix should also be evaluated to avoid a loss of strength. Luo et al. [16] proposed that refined martensite sub-grains (packets, blocks or laths) contributed to the improved impact toughness by acquiring finer retained austenite in an ultra-high-strength stainless steel. Han et al. [17] revealed that much finer prior austenite subjected to multi-stage forging could increase the impact toughness of 18Ni(250) maraging steel. The above indicates that the synergistic toughening of metastable austenite and a hierarchical maraging matrix can effectively resolve the dilemma of inverse strength–toughness. However, there are fewer studies focusing on the above multiple toughening mechanisms in PH13-8Mo steel.

To explore the toughness potential of PH13-8Mo steel to serve as a candidate material in large-scale components bearing an alternating load, it is necessary to elucidate the relationship between impact toughness and heterogeneous microstructures. In this study, subjected to aging at different temperatures, the evolution of a martensite matrix, the distribution of reversed austenite and the precipitation behavior of NiAl particles were observed. Based on the response of the mechanical properties, attention was paid to the toughening mechanisms and their change, according to the above microstructures and post mortem fracture analysis.

## 2. Materials and Methods

Subjected to vacuum induction melting (VIM) + vacuum arc remelting (VAR), the experimental steel was poured into a 50 kg ingot, with the chemical composition given in Table 1. After homogenization at 1150 °C for 6 h, the ingot was forged into a square bar (section: 70 mm × 70 mm) with deformation of −50% in each of three directions (3D). By adopting a thermal dilatometer (Figure 1), the austenite reversion start (Ac1) and finish (Ac3) temperatures were identified as 650 °C and 750 °C, respectively. The precipitation temperature range of the NiAl phases was 430~625 °C, indicated by the enlargement in Figure 1. The martensite transformation start (Ms) temperature was 175 °C. Accordingly, the samples longitudinally cut from the bar were solution-annealed (SA) at 900 °C for 1 h, followed by oil cooling (OC) to room temperature (RT). By referring to previous studies [11,18,19,20,21] to trace the representative conditions of peak aging and over-aging, the samples were subjected to an aging treatment (AT) at 510 °C, 520 °C, 530 °C, 540 °C, 550 °C and 560 °C for 4 h, followed by air cooling (AC). This was a favorable peak aging time to optimize the aging temperature. All the as-aged samples were machined into rod tensile specimens with a gauge diameter of 5 mm and a gauge length of 25 mm. The tensile tests at RT were performed on an Instron 5982 testing machine (Instron; division of ITW Ltd., Cheshire, UK) with a strain rate of 6.7 × 10^−3^ s^−1^. The Charpy 45° V-notch samples (size: 10 mm × 10 mm × 55 mm; depth: 2 mm) were subjected to impact tests at RT on a ZBC2452-C fully automatic impact testing machine (MTS China Holdings LLC, Shenzhen, China) at RT. The average value of the above mechanical properties was obtained by testing three parallel samples; the standard deviation was given accordingly.

The microstructure of the as-aged martensite matrix and impact fracture morphology were observed by an S-3400N field emission scanning electron microscope (SEM) (Hitachi Ltd., Tokyo, Japan). X-ray diffraction (XRD) tests were performed using a Bruker AXS D8 Advance diffractometer (Bruker AXS GmbH, Beijing, China) to identify the constituent phases within the 2θ range of 40~102° and with a scan step interval of 0.02°; the density of dislocation was estimated based on the Williamson–Hall (WH) method [22]. Electron backscatter diffraction (EBSD) maps were collected using an FEI Apreo C SEM (FEI Company, Hillsboro, America) to identify the hierarchical boundaries of the matrix and analyze the impact fractography. The samples were prepared by mechanical polishing and ion milling. Further, the transmission electron microscope (TEM) was used to observe the distribution of reversed austenite and NiAl precipitates with an FEI Tecnai F20 (FEI Company, Hillsboro, America) at an acceleration voltage of 200 kV, equipped with an energy-dispersive X-ray (EDX) system. The thin TEM foils were prepared by TenuPol-5 Struers (Struers, Copenhagen, Denmark) with a solution of 10 vol.% perchloric acid in an alcohol solution at −25 °C and with a current of 35 mA. The complete experimental process is shown in Figure 2.

## 3. Results

### 3.1. Mechanical Properties

Compared with the SA state, the AT-510 sample showed the highest yield strength (YS) of ~1.4 GPa and ultimate tensile strength (UTS) of ~1.5 GPa (Figure 3). There was a continuous softening with the increase in aging temperature. In comparison with Ni_3_Ti-strengthened stainless steels, it should be related to the recovery of the matrix [23], the coarsening of precipitates and the formation of reversed austenite [14,24]. For the AT-560 sample, both of the YS and UTS were the lowest (~1.2 GPa), which were reduced by about 20% compared with the AT-510 sample. The impact energy of AT-550 and AT-560 (~220 J) increased by almost ten times compared with that of AT-510 (~23 J). From the impact fractography, the AT-510 sample (Figure 4a) exhibited a feature of a quasi-cleavage fracture by dominant facets with river patterns and also by the secondary cracks initiated by a pile-up of dislocations. The mixture of cleavage facets and dimples indicated the recovery of impact toughness for the AT-530 sample (Figure 4b) whilst there were more differently sized dimples occupying the fracture surface of the AT-540 sample (Figure 4c). For the AT-560 sample, Figure 4d shows the dimples that became finer and more equiaxial due to the ductile fracture, which was consistent with its highest impact energy. Overall, the improvement of impact toughness was more dominant, despite the inverse relationship of strength−toughness. Both of the AT-540 and AT-550 samples exhibited a desirable combination of a high strength and good toughness.

### 3.2. Martensite Matrix

As shown in Figure 5a, the as-solution annealed sample showed a typical lathy martensite matrix. After aging treatment, for the AT-510 and AT-530 samples (Figure 5b,c), packets with different orientations separated the austenite grains, indicated by image contrast. Subjected to a further recovery of the matrix, the block boundaries in the AT-540 and AT-560 samples (Figure 5d,e) could be identified, accompanied by the growth of martensite sub-grains. In Figure 6a, the XRD results indicated the full martensite in the SA and AT-510~540 samples. There was a weak (111)_γ_ peak in the AT-560 sample, shown in the inset, indicating the limited austenite reversion. Further, Figure 6b shows the largest density of dislocation in the SA sample (~3.35 × 10^15^); it continued decreasing with the increase of aging temperature due to the recovery of the martensite matrix. However, it further recovered in the AT-560 sample, possibly due to the formation of more sub-grain boundaries. Figure 7 shows the evolution of the hierarchical martensite matrix, decorated by high-angle grain boundaries (HAGBs) over 15°. There was no austenite found in the as-solution annealed and as-aged matrixes, which was consistent with the above XRD results. Due to the recovery of the matrix after aging, the equivalent diameter of the as-aged PAGs was lower than the as-solution annealed sample; further increasing the aging temperature led to a larger size of PAGs (23.8 μm), as shown in Figure 7f. By comparison, the distribution of block boundaries became densest within packets for AT-540 among all samples, whilst the average misorientation and block size were determined as ~13.8° and ~2.26 μm, respectively (Figure 8). Similarly, there was a growth of martensite sub-grains with the increase in aging temperature.

### 3.3. NiAl Precipitates and Reversed Austenite

Figure 9 reveals that the evolution of the martensite matrix was precipitated by nano-precipitates. As shown in Figure 9a, the AT-510 sample exhibited fully aged martensite from the BF image. By adopting the spots of secondary diffraction from the insertion, there was a high density of globular precipitates distributed in the matrix from the DF image (Figure 9(a1,b1)). The precipitates could be determined as ordered B2-CsCl NiAl with a body-centered cubic (BCC) crystal structure, based on previous studies [26,27]. It exhibited a coherent orientation relationship with the martensite matrix from the <001>_α_ zone axis. For the AT-540 sample, the NiAl precipitates grew up in the matrix, maintaining a high density of dislocations (Figure 9(b1,b2)). Subjected to a higher aging temperature of 560 °C, it was found that the NiAl precipitates grew further, yet without coarsening, in the recovered matrix, as shown in Figure 9(c1,c2). Using Gaussian fitting, Figure 10 shows the distribution of precipitates in terms of the diameter and inter-particle distance. The maximum value of the inter-particle distance was below 30 nm whereas the diameter was within the range of 2~8 nm, consistent with the previous experimental and predicated results [8].

By adopting the STEM-HAADF image, Figure 11a indicates the typical lathy martensite matrix with a high density of dislocations in the AT-510 sample. As the aging temperature increased to 540 °C, there was a further coalescence of martensite laths and filmy reversed austenite formed along the lath boundaries, as shown in Figure 11b. The SAED pattern showed a Kurdjumov−Sachs (K−S) orientation of ${\left[\bar{1}11\right]}_{\alpha }//{\left[011\right]}_{\gamma }$ and ${\left(110\right)}_{\alpha }//{\left(11\bar{1}\right)}_{\gamma }$ between the austenite and martensite. For the AT-560 sample (Figure 11c), filmy reversed austenite grew and separated the matrix; NiAl precipitates could still be detected, based on the SAED pattern (Figure 11d). Compared with the AT-540 sample, it was found that the average width increased from ~66 nm to ~86 nm and the length increased from ~340 nm to ~410 nm, respectively. Further, the EDX mapping (Figure 11e) indicated the enrichment of Ni in the reversed austenite and the local partitioning of austenite stabilizer elements contributed to the reversion of the matrix. The above indicated that nano-scale austenite had already formed, but could not accurately be detected by XRD.

## 4. Discussion

### 4.1. Toughening Mechanisms

With the increase in aging temperature, the impact toughness improved from the peak aging of AT-510 to the over-aging of AT-540 and AT-560, without much sacrifice of tensile strength. The variation in toughening mechanisms was discussed based on the microstructural evolution, as follows.

#### 4.1.1. Toughening from the Matrix

For ultra-high-strength steel, a hierarchically structured martensite matrix is one of the main factors controlling toughness [16]. Refined martensite sub-grains and an increased ratio of HAGBs with an increase in aging temperature were noticed, as shown in Figure 7. Previous studies have indicated the possibility of martensite sub-structures of laths, blocks and packets serving as toughening units [16,25,28]. To identify the controlling type of sub-structure, a post mortem EBSD analysis was conducted on a cross-sectional impact fracture, as shown in Figure 12. Along the main cracking direction, more flat cracks (such as P1~P3) directly penetrated the martensite laths with LAGBs in the AT-510 sample (Figure 12a) whereas a more frequent deflection was found with the HAGBs from the rectangle regions. By measuring the angle where deflection occurred, Table 2 shows that the ratio of HAGBs (63%) outweighed the lath boundaries (37%). The above indicated that the HAGBs could serve as an effective toughening unit to increase the resistance of crack propagation in the AT-510 sample.

By comparison, there were more zigzag crack paths in the AT-540 sample (Figure 12b) and most of deflection was located at the inter-laths, defined by LAGBs from the noted regions. Table 2 shows a higher ratio of 58% for the lath boundaries. In addition to the recovered matrix (Figure 6b) and the refined blocks (Figure 8b), the laths should have been more dominant in the toughening mechanisms. This was closely related to the reversed austenite formed at the lath boundaries (Figure 11b), serving as “soft barriers” [10] to prevent direct penetration across the block units. Given the improved ratio of HAGBs in Figure 8a, the possible increased misorientation between laths may also change the advance direction and increase the crack paths by resisting the continuous intra-grain gliding of mobile dislocations. In this sense, the recovered laths embedded by soft austenite contributed to the improvement in impact toughness, indicated by the fine and high density of the dimples (Figure 4c).

#### 4.1.2. Toughening from Reversed Austenite

Figure 4d indicates a complete ductile fracture in the AT-560 sample with a much higher impact energy of 220 J. Considering the decreased ratio of HAGBs and increased size of blocks in Figure 8, there should have been a change in the toughening mechanism from the martensite matrix to reversed austenite with an aging temperature increase above 540 °C. The segregation of Ni at lath boundaries [10] or the local dissolution of Ni_3_Ti precipitates [29] can promote the nucleation and growth of reversed austenite. Initially, filmy austenite with a width below 100 nm could be found in the AT-540 sample (Figure 11b); it grew up along the lath boundaries as the aging temperature increased to 560 °C. The EDX maps showed the enrichment of Ni and Cr elements to yield the high thermality of reversed austenite [30], which could stably be reversed at room temperature. Previous studies [16,31,32,33] have paid much attention to the relationship between the mechanical stability of austenite and the toughening effect. Although it has been found that smaller austenite is less stable under tensile loads [31], fine filmy austenite contributes to the work hardening [32] and traps the propagating cracks [16] under an impact load. Our studies on Fe-10Cr-10Ni stainless steel found that larger blocky reversed austenite cannot further improve the toughness due to the low mechanical stability [14]. Comparatively, there was more predominantly filmy reversed austenite in a higher amount in the AT-560 samples. In addition to the above “soft barrier” effect, the transformation-induced plasticity (TRIP) effect can introduce residual compressive stress from in situ austenite-to-martensite transformations [34], which can blunt the crack tips to resist propagation.

#### 4.1.3. Distribution Impact of NiAl Precipitates

It is known that the high density of precipitates contributes to an ultra-high strength; it also has an adverse impact on ductility or toughness due to the limited work-hardening capacity [35,36] of the maraging matrix. In maraging steels strengthened by Ni_3_Ti precipitates, there is a tendency of coarsening after increasing the aging temperature, which leads to the embrittlement of the maraging matrix due to the loss of a coherent interface. By comparison, the impact toughness was continuously improved without much softening (Figure 3). There was a well-coherent interface between NiAl and martensite as the SAED indicated (Figure 9(a2–c2)). The calculated lattice misfit was only about 0.72% between the stoichiometric NiAl and the ferrite matrix, based on [4]. This could prevent the coarsening of NiAl by controlling its size when given low interfacial energy, according to the Lifshitz, Slyozov and Wagner (LSW) theory [37,38]. The resultant average diameter of NiAl particles was below 10 nm for all as-aged samples. Meanwhile, a well-coherent interface could provide compatible deformation by the cutting of lattice dislocations. As [39] indicated, the associated elastic interaction between NiAl and the matrix should be lowered in terms of negligible strain accumulation, which would delay the pile-up of dislocations before the NiAl particles. According to previous studies on 18Ni(250) steel [40], the distance between the crack tips and nucleation sites of microvoids increases with the inter-particle distance. It could be speculated that the larger inter-particle distance of NiAl precipitates (Figure 10) was also beneficial to the impact toughness by increasing the propagation energy of cracks. From the above analysis, the distribution of NiAl precipitates in terms of their limited nano-size and increased inter-particle distance provided a favorable foundation for toughness by avoiding the embrittlement of the matrix after over-aging.

### 4.2. Variation in Toughening Mechanisms

According to the response of the mechanical properties, the toughening mechanisms could be divided into three stages. The following pays attention to the variations, based on the schematics in Figure 13.

Stage I: Corresponding with low-temperature aging around 510 °C (Figure 13a). Although there was a peak aging strength of ~1.5 GPa, the maraging matrix with a high density of dislocation and NiAl precipitates could easily lead to the pile-up and tangle of dislocations and further the initiation of cracks. However, compared with the laths directly penetrated by cracks, the hierarchical sub-grains with HAGBs could retard the propagation of cracks to obtain a certain impact toughness, resulting in the quasi-cleavage fracture in Figure 4a.

Stage II: In the case of intermediate-temperature aging around 540 °C, the toughening was dominated by the recovered and refined martensite matrix (Figure 13b). Due to the decreased density of dislocation inherited from the solution-annealed matrix, the interaction between the mobile dislocation and NiAl particles could be enhanced to release the rate of dislocation pile-up and the nucleation of cracks. After considering the increased ratio of HAGBs, the main toughening mechanism originated from the intra-granular laths glued by the reversed austenite (Figure 13b), leading to increased resistance on the cracks both from increasing the advance path and the blunting effect from the TRIP effect.

Stage III: For high-temperature aging around 560 °C, Ni-enriched nano-scale reversed austenite could blunt the tips of cracks and resist penetration between the martensite sub-grains. The TRIP effect could also mostly absorb the impact energy (Figure 13c). These beneficial factors offset the weakness of a decreased ratio of HAGBs and increased size of blocks. It should be emphasized there was a well-matched coherent interface between NiAl and the matrix; its intrinsic resistance ability of coarsening and intra-granular precipitation simultaneously impeded the sacrifice of the strength–toughness of the matrix.

## 5. Conclusions

A better combination of strength and impact toughness could be achieved after aging treatments between 540 and 550 °C. Compared with the highest strength of aging at 510 °C (1.5 GPa), the impact energy could be increased by almost ten times to reach 220 J, whilst still maintaining a desirable yield strength of 1.3 GPa.With an increase in aging temperature, there was a continuous recovery of the matrix. When increasing the aging temperature to 540 °C, it acquired the lowest density of dislocations (~2.73 × 10^14^) and most refined effective grain (~2.26 μm); after aging at 560 °C, there were Ni-enriched austenite films and a greater coalescence of martensite laths without the coarsening of NiAl precipitates (average diameter of 8.58 nm and inter-particle distance of 27.8 nm).There were multiple toughening mechanisms as a function of aging temperature. Stage I: for low-temperature aging, the HAGBs contributed to the toughness by retarding the advance of cracks. Stage II: for intermediate-temperature aging, the recovered laths embedded by soft austenite mostly facilitated the recovery of toughness from increasing advance paths and blunting the tips of cracks. Stage III: for high-temperature aging, given the stable NiAl precipitates, more inter-lath reversed austenite led to the optimum toughness, relying on the “soft barrier” and TRIP effects.

## Figures and Tables

**Figure 1 materials-16-03630-f001:**
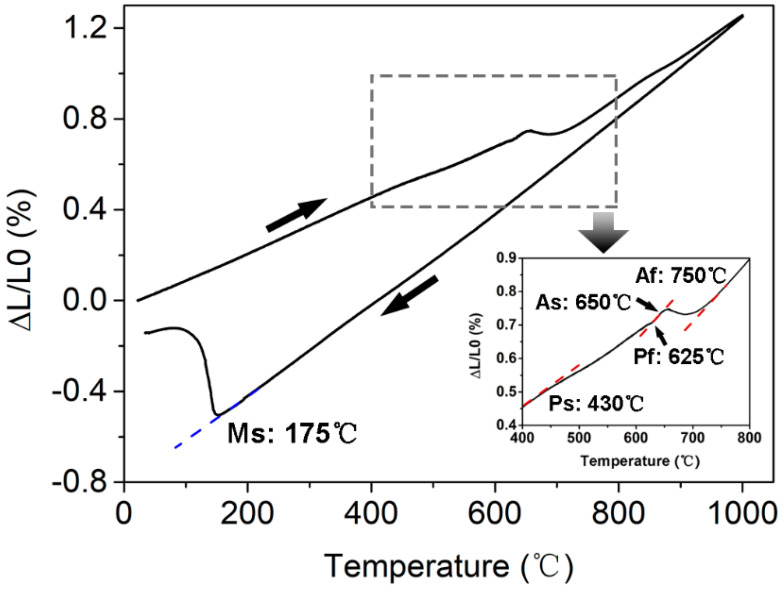
The dilatometric curve indicating the phase transformation temperatures of the experimental steel.

**Figure 2 materials-16-03630-f002:**
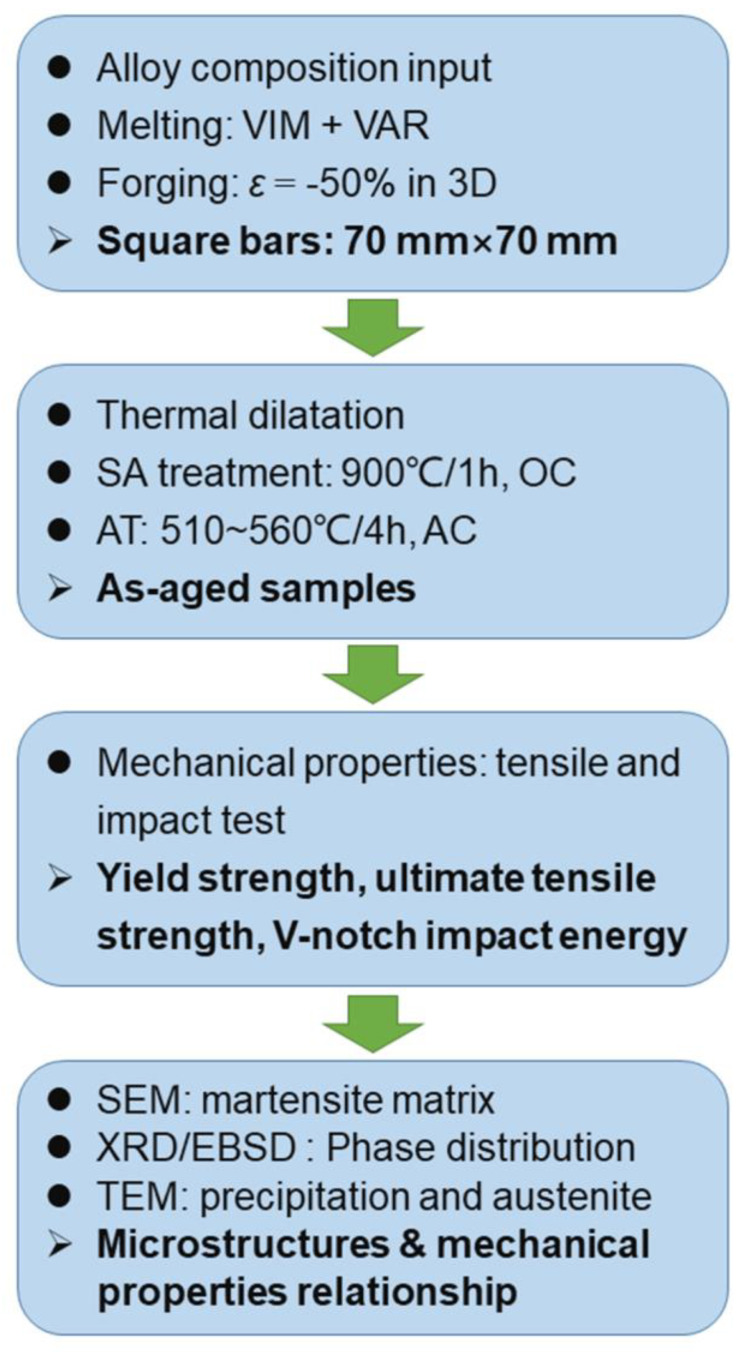
Experimental process flow chart.

**Figure 3 materials-16-03630-f003:**
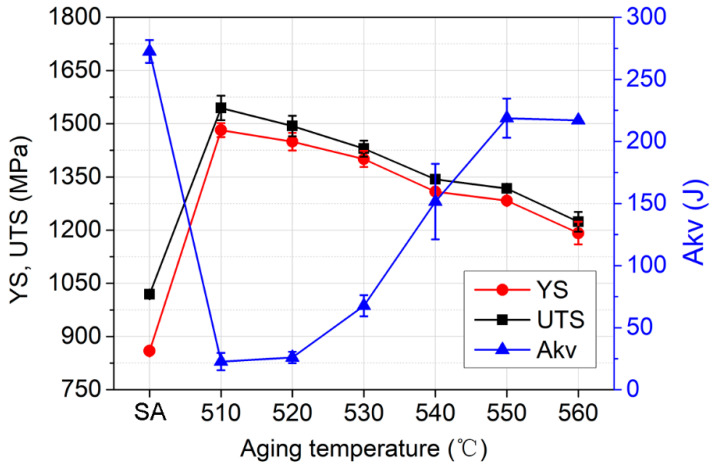
Tensile strength (YS and UTS) and impact toughness (Akv) of the solution-annealed (SA) and as-aged samples.

**Figure 4 materials-16-03630-f004:**
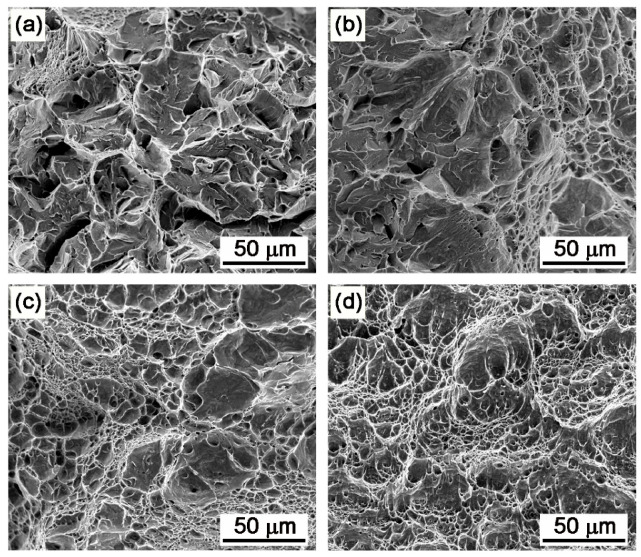
Impact fracture morphology of the samples under different aging temperatures: (**a**) AT-510; (**b**) AT-530; (**c**) AT-540; and (**d**) AT-560.

**Figure 5 materials-16-03630-f005:**
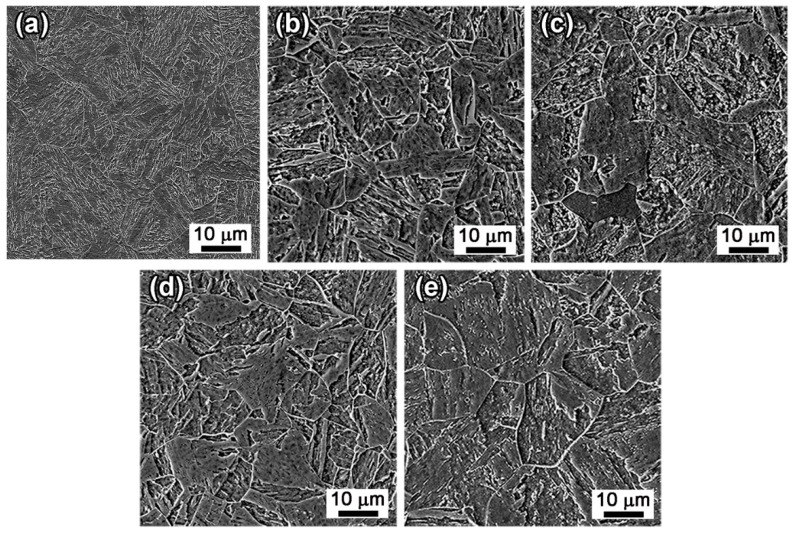
SEM images of the (**a**) as-solution annealed, (**b**) AT-510, (**c**) AT-530, (**d**) AT-540 and (**e**) AT-560 samples.

**Figure 6 materials-16-03630-f006:**
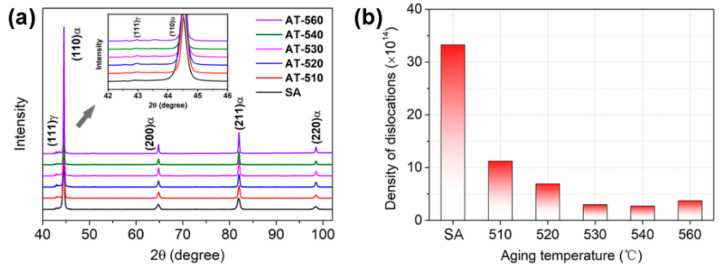
(**a**) XRD patterns of the as-solution annealed and as-aged samples, wherein the inset corresponds with the enlargement of (111)_γ_ peak. (**b**) Density of dislocations as a function of aging temperature.

**Figure 7 materials-16-03630-f007:**
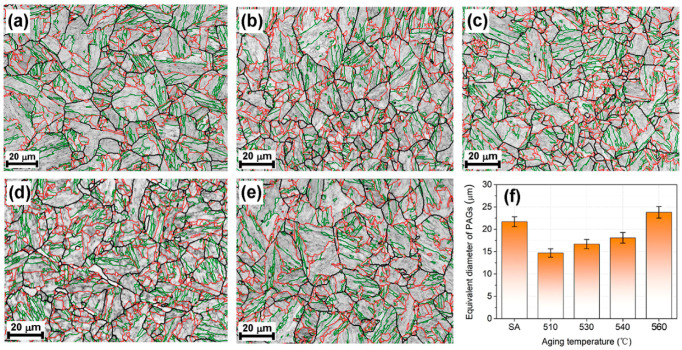
EBSD image quality (IQ) phase maps with misorientation distribution of martensite grain boundaries of the (**a**) as-solution annealed, (**b**) AT-510, (**c**) AT-530, (**d**) AT-540 and (**e**) AT-560 samples; (**f**) equivalent diameter of prior austenite grains (PAGs). The green, red and black lines correspond with the blocks, packets and prior austenite boundaries, respectively (all HAGBs over 15°).

**Figure 8 materials-16-03630-f008:**
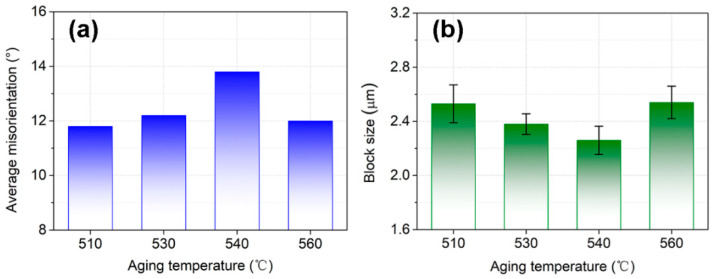
Variations in (**a**) average misorientation and (**b**) block size of the martensite as a function of aging temperature, which were obtained based on our previous methods in [25].

**Figure 9 materials-16-03630-f009:**
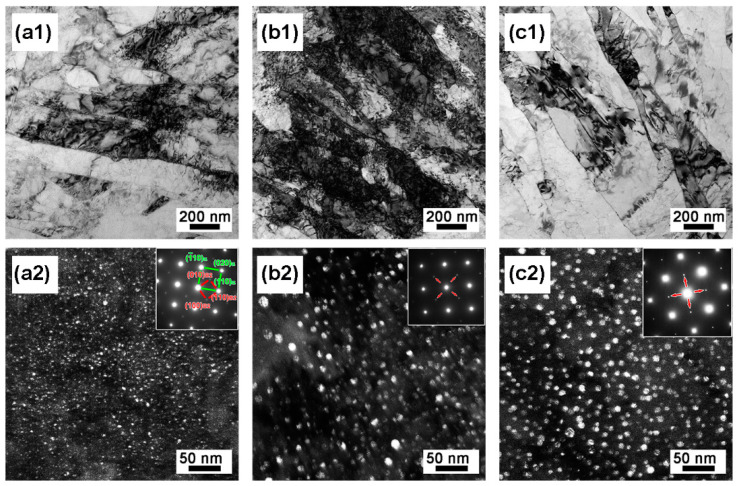
(**a1**–**c1**) Bright-field TEM (BF-TEM) image and (**a2**–**c2**) dark-field TEM (DF-TEM) image of (**a1**,**a2**) AT-510, (**b1**,**b2**) AT-540 and (**c1**,**c2**) AT-560 samples.

**Figure 10 materials-16-03630-f010:**
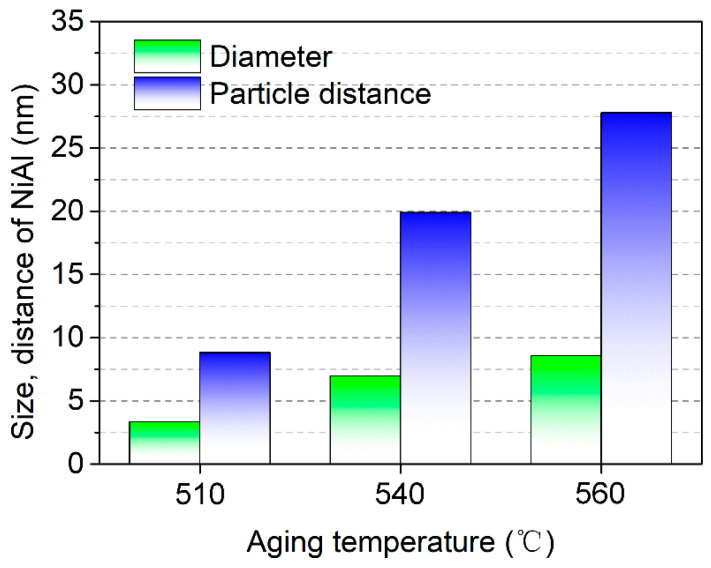
Variations in the size and inter-particle distance of NiAl precipitates as a function of aging temperature, which were obtained based on our previous methods in [25].

**Figure 11 materials-16-03630-f011:**
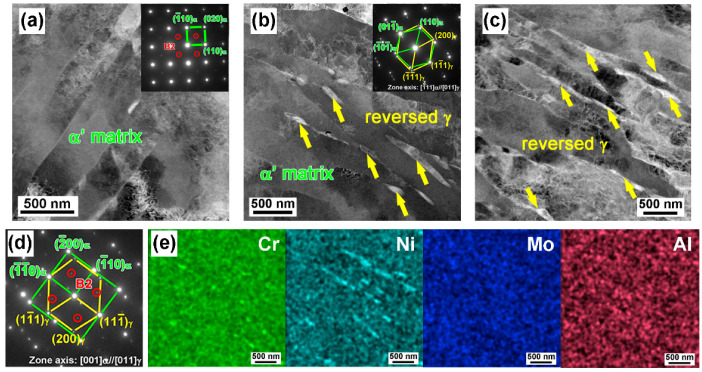
Scanning TEM high-angle annular dark-field (STEM-HAADF) image of the microstructure of (**a**) AT-510, (**b**) AT-540 and (**c**) AT-560 samples. (**d**) Selected area electron diffraction (SAED) pattern showing the orientation relationship of NiAl/austenite with matrix. (**e**) Energy-dispersive X-ray (EDX) maps showing the main elemental composition of the region in (**c**).

**Figure 12 materials-16-03630-f012:**
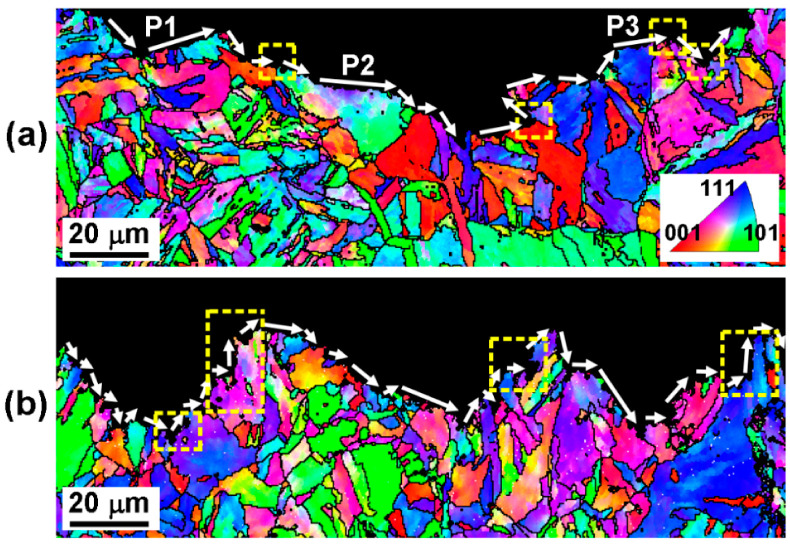
Inverse pole figure (IPF) map of the cross-sectional region close to the fracture surface of (**a**) AT-510 and (**b**) AT-540 samples, wherein the arrows indicate the propagation direction of cracks. The black line indicates HAGBs over 15°.

**Figure 13 materials-16-03630-f013:**
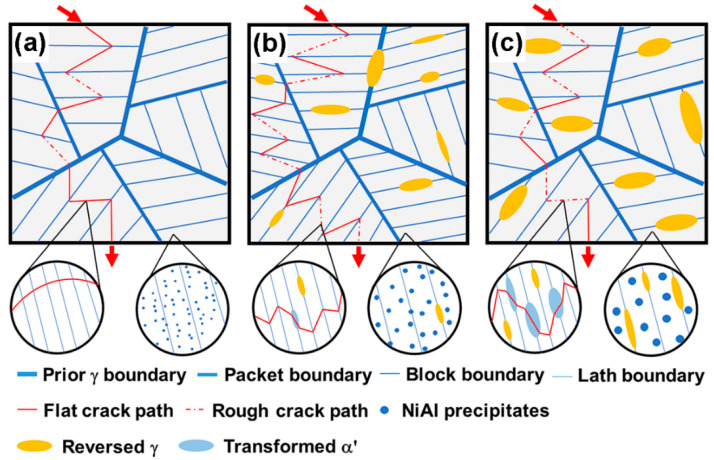
Schematic diagram showing the crack propagation and corresponding evolution of microstructures for (**a**) AT-510, (**b**) AT-540 and (**c**) AT-560 samples.

**Table 1 materials-16-03630-t001:** The chemical composition of the experimental steel (wt.%).

C	Cr	Ni	Mo	Al	Si	Mn
0.018	12.72	8.24	2.33	1.20	0.02	0.01

**Table 2 materials-16-03630-t002:** The statistical ratio of boundary types where crack deflection occurred along the crack paths.

Sample	Lath	Packet	Block	Prior Austenite
AT-510	37%	10%	33%	20%
AT-540	58%	15%	14%	14%

## Data Availability

The data presented in this study are available on request from the corresponding author.

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
