# Peer review of "Evolution of Toughening Mechanisms in PH13-8Mo Stainless Steel during Aging Treatment"

_materials, 2023, doi:10.3390/ma16103630_

Round 1

Reviewer 1 Report

All the comments are indicated in the pdf file. The comments are in red

Reviewer 2 Report

The manuscript describes the investigation of the evolution of toughening mechanisms in PH13-8Mo stainless steel as a function of aging temperature.

The topic is novel and of practical significance, and the methods are appropriate. The results are well presented.

However, the reproducibility is compromised by the fact that:

1. No detailed experimental procedure of each test was provided, nor was the full product information (serial No., brand, manufacturer, city, country) of each instrument used.

2. How misorientation distribution, martensite block size, size and inter-particle distance of NiAl precipitates were determined is not clear.

3. EDX result has been presented but the measurement was not introduced in Methods.

There also lack statistical analyses to verify the statement “…the impact toughness remarkably improves from the peak-aging of AT-510 to the over-aging of AT-540 and AT-560, without obvious sacrifice of tensile strength.”

English usage needs to be improved.

Thus a major revision is recommended before further consideration.

Some suggestions:

1. Is Fig. 1 a result? If yes, it should be presented in result section.

2. L109, should be by about 20%

3. Fig. 2 caption should state which sample.

4. remarkably, obvious and such should not be used in scientific works.

5. “…systematic investigation was paid on the evolution of toughening mechanisms in PH13-8Mo stainless steel as a function of aging temperature” should be changed to …investigation of the evolution of…temperature was carried out.

6. Who proposed? It (who is it?) or you? Or you meant to write it was proposed that…?

it observed?

7. “It is known that the high-density of precipitates contribute to the ultra-high strength, while it also has an adverse impact on the ductility or toughness due to the limited work hardening capacity of the maraging matrix.” needs references. This reviewer suggests citing Tian, Kun V., et al. "Composition―Nanostructure Steered Performance Predictions in Steel Wires." Nanomaterials 9.8 (2019): 1119; Tian, Kun V., et al. "Orthodontic archwire composition and phase analyses by neutron spectroscopy." Dental Materials Journal 36.3 (2017): 282-288.

moderate English editing is needed.

Round 2

Reviewer 2 Report

All points have been addressed and the manuscript is ready for publication.